# Optimization of Transcardiac Perfusion for More Accurately Evaluating Biodistribution of Large Molecules

**DOI:** 10.3390/ijms252212180

**Published:** 2024-11-13

**Authors:** Zuoxu Xie, Annie Guo, Ekta Kadakia

**Affiliations:** Drug Metabolism and Pharmacokinetics, Biogen, Cambridge, MA 02142, USA; zuoxu.xie@biogen.com (Z.X.); annie.guo@biogen.com (A.G.)

**Keywords:** transcardiac perfusion, sodium nitrite, fluorescent dextran, residual blood

## Abstract

The accurate assessment of drug concentrations in biodistribution studies is crucial for evaluating the efficacy and toxicity of compounds in drug development. As the concentration of biologics in plasma can be higher than in tissue due to their potentially low volume of distribution, transcardiac perfusion is commonly employed to reduce the influence of excess drugs in residual blood. However, there is a lack of consistency in the literature on the conditions and methods of perfusion. To enhance blood removal during transcardiac perfusion, sodium nitrite (NaNO_2_), a vasodilator, has been widely used with concentrations up to 5% in publications. However, we found that such high NaNO_2_ could disrupt the BBB during perfusion, which should be avoided in experiments. In this study, we examined the impact of various vasodilators on blood–brain barrier integrity and vascular permeability using the ratio of FITC-Dextran to Texas Red-Dextran (FITC/Texas Red). Additionally, we optimized perfusion conditions—including euthanasia method and perfusion flow rate—based on hemoglobin levels and the FITC/Texas Red ratio in tissues. Despite the superiority of NaNO_2_ in terms of solubility and cost over other vasodilators, we found that 2% NaNO_2_ disrupted blood–brain barrier integrity, significantly altering the FITC/Texas Red ratio. In contrast, 100 mM NaNO_2_ did not significantly affect this ratio. Moreover, under Ketamine/Xylazine (Ket/Xyl) anesthesia, which reduced blood clot formation compared to CO_2_ euthanasia, 100 mM NaNO_2_ achieved the lowest hemoglobin levels in the brain. Compared to other vasodilators and the PBS control group, 100 mM NaNO_2_ decreased the tissue/plasma ratio (K_p,t_) but not brain/plasma ratio (K_p,b_) of hIgG1 and human transferrin. We have developed a method to efficiently evaluate blood–brain barrier integrity during transcardiac perfusion. The combination of Ket/Xyl anesthesia and 100 mM NaNO_2_ effectively removes residual blood from tissues without significantly affecting blood vessel permeability.

## 1. Introduction

In drug development, accurately assessing drug concentrations in tissues is crucial for predicting both efficacy and toxicity. Although various in vitro and in silico methods exist to predict tissue drug concentrations [1,2], the most reliable measure is the total or unbound tissue/plasma partition coefficient (K_p,t_ or K_p,uu,t_) obtained from in vivo experiments. This method directly measures drug concentrations in tissue and plasma [3]. However, the total tissue concentration (C_t_) obtained from in vivo experiments does not represent the exact drug concentration in the tissue; instead, it is the sum of the vascular contribution (A_pla_, the drug remaining in blood vessels) and the tissue extravascular contribution (A_ext_), as shown in Equation (1). Therefore, to accurately estimate K_p,t_, it is essential to consider contamination from the drug in residual blood. This is particularly important for biologics, which are mainly restricted to the vascular space and have a low volume of distribution [4], leading to higher drug concentrations in plasma than in tissues, especially in the brain.
C_t_ = (A_pla_ + A_ext_)/V_t_(1)

To reduce the influence of residual blood on quantifying drug concentration in tissues, transcardiac perfusion is commonly employed in rodent models. However, the literature lacks consistency in reporting perfusion conditions and methods, which vary considerably in terms of perfusion volume, flow rate, and perfusate composition. For instance, Carl I. Webster et al. perfused rats with 20 mL of heparinized (1 EU/mL) saline at 2 mL/min [5], while Kirk W. Johnson et al. simply mentioned performing perfusion to rinse out residual blood [6]. Regarding mice, Shah’s group used 8–10 mL of PBS at a rate of 10–20 mL/min [7], while Syvänen’s group perfused mice with 40 mL of 0.9% NaCl in 2.5 min [8]. Sodium nitrite (NaNO_2_), a vasodilator, has also been widely used to enhance blood removal during transcardiac perfusion, with concentrations varying significantly between 0.0002% and 5% in different publications [9,10,11,12].

Given the variability in perfusion methods and the lack of thorough investigation into their effectiveness for blood removal, our department conducted a series of experiments to assess various perfusion conditions by measuring hemoglobin levels in the brain. In previous studies, we found that a 20 mins perfusion with 2% NaNO_2_ significantly improved blood removal in mice compared to PBS perfusion, but not in rats. Although we claimed that 2% NaNO_2_ perfusion does not disrupt the integrity of the blood–brain barrier (BBB) in mice, we lacked the tools to evaluate BBB integrity during perfusion [13]. Additionally, the effectiveness of transcardiac perfusion in removing blood from other tissues had not been previously investigated. Therefore, further optimization of the perfusion method is needed, along with an evaluation of its impact on blood vessel permeability, particularly in the BBB.

Agents such as FITC-Dextran, albumin, Evans Blue, sodium fluorescein, and IgG are commonly used to assess alterations in BBB permeability [14]. The choice of a suitable agent is critical, as the marker must remain within healthy cerebral blood vessels but penetrate damaged ones. Small molecules like Evans Blue and sodium fluorescein are not ideal candidates; although Evans Blue can bind to albumin to form a larger molecule, unbound Evans Blue may still diffuse through an intact BBB [15]. Albumin and IgG are large molecules, but their quantification is costly and time-consuming, and their uptake by the liver may affect the interpretation of data [16,17]. Dextrans are polysaccharides composed of glucose molecules, with molecular weights ranging from 4 to 200 kDa. They are commercially available with various conjugated fluorescent probes, which are essential for determining the BBB integrity during vasodilator perfusion. An added benefit of fluorescent dextran is that its concentration can be quantified simply and quickly using a fluorescence plate reader. Therefore, we have decided to use fluorescent dextran to determine BBB integrity.

As shown in Figure 1A, the concentration of FITC-Dextran in the brain includes contributions from both vascular and extravascular FITC-Dextran (Equation (1)). Perfusing FITC-Dextran alone cannot distinguish between these two scenarios in the vasodilator group. Due to the dilation of blood vessels caused by vasodilators, both scenarios could increase the FITC-Dextran concentration in the vasodilator group from two to six compared to the PBS group. In scenario 1, the vasodilator disrupts the BBB, increasing both vascular and extravascular FITC-Dextran contributions. In scenario 2, the vasodilator only increases vascular FITC-Dextran without affecting BBB integrity. To address this, we introduced another fluorescent dextran, Texas Red-Dextran, as a blood vessel volume marker. As described in Figure 1B, after perfusing the mouse with FITC-Dextran for a set duration, Texas Red-Dextran is added to the perfusate in the final minutes. Assuming Texas Red-Dextran does not cross blood vessels and accumulate in the tissue within this short time, the ratio of FITC-Dextran to Texas Red-Dextran (FITC/Texas Red) can be used to determine BBB integrity. If the BBB is disrupted by the vasodilator during perfusion, the FITC/Texas Red ratio will increase from one to two. In contrast, the ratio will remain unchanged if the BBB is intact.

In this work, we adopted a new approach by perfusing FITC-Dextran and Texas Red-Dextran successively to evaluate BBB integrity during vasodilator perfusion. Using the FITC/Texas Red ratio generated from this approach, along with hemoglobin levels in the tissue, we assessed the impact and performance of various vasodilators on blood vessel permeability and the removal of residual blood in the brain. After optimizing the perfusion procedure, we further investigated the effectiveness of vasodilator perfusion on blood removal and its influence on vascular permeability and drug concentration in other tissues, including the liver, kidney, and spleen. Thoroughly evaluating the impact of vasodilators on perfusion efficiency and vascular permeability is critical for selecting the appropriate perfusion method in biodistribution studies to accurately determine the concentrations of large molecules in target tissues.

## 2. Results

### 2.1. Effect of 2% NaNO_2_ on Blood Removal and BBB Integrity

Hemoglobin concentration in the tissue was used as a biomarker to indicate the presence of residual blood. Adding 2% NaNO_2_ significantly improved the effectiveness of PBS perfusion in clearing blood from the brain, reducing the relative hemoglobin concentration from 19.8% to 0.3% (Figure 2A). To assess the impact of perfusion on the brain drug distribution, the K_p,b_ of hIgG1 was measured 24 h after intravenous injection, with or without perfusion. PBS perfusion notably decreased the K_p,b_ of hIgG1 to 0.0028. Furthermore, adding 2% NaNO_2_ to the perfusion solution reduced the K_p,b_ value by an additional 50% (Figure 2B).

While 2% NaNO_2_ demonstrated significant efficacy in blood removal, we also needed to evaluate its impact on BBB integrity during perfusion. As described in the Methods section, we compared the FITC-Dextran concentration and the ratio of FITC-Dextran to Texas Red-Dextran in the brain between 2% NaNO_2_ and PBS perfusions to assess BBB integrity. Figure 2C,D show that 2% NaNO_2_ perfusion significantly increased both FITC-Dextran concentration and the FITC/Texas Red ratio in the brain compared to PBS perfusion. The substantial difference (18-fold) in the FITC/Texas Red ratio between the 2% NaNO_2_ group and the PBS group indicates BBB disruption during 2% NaNO_2_ perfusion.

### 2.2. Evaluation of the Performance of Vasodilators

In addition to 2% NaNO_2_, we tested three other vasodilators (bradykinin, papaverine, and nimodipine) and two different concentrations of NaNO_2_ (1 mM and 100 mM). While these alternatives also reduced the hemoglobin concentration in the brain compared to PBS, none were as effective as 2% NaNO_2_. The highest-performing alternative, 100 mM NaNO_2_, decreased the remaining blood to 1.4% (Figure 3A). However, unlike 2% NaNO_2_, the FITC-Dextran concentration in the brain did not increase significantly for the other perfusion groups.

Figure 3B shows that, compared to the PBS group, 100 mM NaNO_2_ increased FITC-Dextran concentration in the brain. However, Texas Red-Dextran concentration was also higher in the 100 mM NaNO_2_ group, resulting in an FITC/Texas Red ratio similar to that of the PBS group. This indicates that BBB integrity was maintained with 100 mM NaNO_2_. To confirm these results, mice were perfused with an Evans Blue in 1% BSA solution followed by 2 min of PBS perfusion. According to Figure 3E, the brain was stained blue in the 2% NaNO_2_ group but not in the PBS or 100 mM NaNO_2_ groups.

### 2.3. Optimization of Perfusion Procedure

Before perfusion, mice need to be euthanized or anesthetized. As shown in Figure 4A, using Ket/Xyl anesthesia significantly reduced hemoglobin concentration in the brain compared to CO_2_ euthanasia prior to PBS perfusion. Additionally, there was no significant difference in the FITC/Texas Red ratio between these two methods (Figure 4B). During perfusion, we found that the perfusion rate affected the effectiveness of PBS perfusion. Although no statistical difference was observed between groups, as shown in Figure 4C, a rate of 10 mL/min demonstrated the best blood removal efficacy, decreasing hemoglobin concentration in the brain to 1.25%.

We also investigated the influence of osmotic pressure on the performance of NaNO_2_ perfusion. Adding NaNO_2_ directly into the isotonic PBS solution increased the osmotic pressure from 283 to 463 mOsmol/L, making it hypertonic. To study this influence, we prepared isotonic and hypertonic 100 mM NaNO_2_ solutions for perfusion. The hypertonic 100 mM NaNO_2_ solution removed more hemoglobin from the brain vessels (Figure 4D).

### 2.4. Hemoglobin Concentration in Different Tissues

Based on our results, Ket/Xyl anesthesia and a 10 mL/min perfusion rate were effective for flushing blood out of the mouse brain. We then verified the effect of various vasodilators on blood removal from brain tissue using this optimized perfusion procedure. Additionally, we investigated the effectiveness of vasodilators in clearing blood from other organs, including the liver, kidney, and spleen.

As expected, 100 mM NaNO_2_ was the most effective at removing blood from the cerebral vasculature in mice (Figure 5A). The hemoglobin concentration in the groups treated with the other two vasodilators was similar to that of the PBS group. Similarly, 100 mM NaNO_2_ had the lowest hemoglobin concentration in the kidney compared to the nimodipine and bradykinin groups, which were comparable to the PBS group (Figure 5C). This phenomenon was not observed in the liver and spleen tissue. As shown in Figure 5B, there was no difference in liver hemoglobin concentration, which ranged from 1.41% to 1.86% across all four groups. In the spleen, the hemoglobin concentration was extremely high in all groups (33.3–77.8%, Figure 5D).

### 2.5. Fluorescent Dextrans Concentration in Different Tissues

In Section 2.1 and Section 2.2, we demonstrated that FITC-Dextran and Texas Red-Dextran could be used to determine the integrity of the BBB after mice were euthanized with CO_2_. Using this method, we measured FITC-Dextran and Texas Red-Dextran concentrations in different tissues after replacing CO_2_ euthanasia with Ket/Xyl anesthesia.

For brain tissue, although FITC-Dextran concentration increased in the 100 mM NaNO_2_ group compared to the PBS group, Texas Red-Dextran concentration was also higher in the 100 mM sodium nitrite group (Figure 6A,B). As a result, there was no difference in the ratio of FITC-Dextran to Texas Red-Dextran between the groups (Figure 6C). Similarly, in the liver and spleen, both FITC-Dextran and Texas Red-Dextran concentrations were higher when mice were perfused with 100 mM NaNO_2_ compared to PBS (Figure 6D,E,J,K). Consequently, the FITC/Texas Red ratio showed only minor differences between the PBS and 100 mM NaNO_2_ groups. Regarding the kidney, in Figure 6G–I, we found an increase in Texas Red-Dextran but not FITC-Dextran in the 100 mM s NaNO_2_ group, resulting in a lower FITC/Texas Red ratio in the NaNO_2_ group (0.65) compared to the PBS group (1.35).

### 2.6. Biodistribution of hIgG1 and hTf

To investigate the influence of perfusion on the quantification of large molecules in biodistribution studies, we injected hIgG1 and hTf 24 h prior to perfusion and analyzed protein concentrations in different tissues. Compared to no perfusion, all perfusion groups significantly reduced hIgG1 and hTf concentrations in all tissues. However, there was no significant difference in the K_p,b_ of hIgG1 and hTf among the vasodilators and PBS perfusion groups (Figure 7A,B), even though NaNO_2_ further decreased the hemoglobin level in the brain to 0.16% compared to PBS perfusion (4.13%).

Similarly, the concentration of hIgG1 and hTf in the liver after perfusion did not change with the addition of vasodilators (Figure 7C,D). However, 100 mM NaNO_2_ perfusion did reduce the K_p,t_ of hIgG1 and hTf in kidney and spleen tissues compared to PBS perfusion. As shown in Figure 7E,G, the K_p,t_ of hIgG1 in the 100 mM NaNO_2_ group was lowest in both kidney (0.0068) and spleen (0.0094) tissues. For the K_p,t_ of hTf, although the values in the 100 mM NaNO_2_ group were relatively low, there was no significant difference between the groups.

## 3. Discussion

The development of brain-penetrating biologics and ‘brain shuttle’ technologies has recently gained tremendous importance, particularly with the approval of biologics such as Lecanemab and Donanemab in the United States and pabinafusp alfa in Japan [18,19,20]. Accurate assessment of brain concentrations is essential for advancing these biologics and informing the development of novel platforms and strategies. To obtain precise measurements of biologic concentrations in the brain from in-vivo experiments, transcardiac perfusion is routinely employed to minimize contamination from residual blood in the cerebral vasculature. To enhance the effectiveness of perfusion, researchers often incorporate vasodilators, such as NaNO_2_, into the perfusate to facilitate blood vessel dilation. While our team has demonstrated that 2% NaNO_2_ in PBS effectively removes blood from the cerebral vasculature, we lacked a reliable method to evaluate the integrity of the BBB during perfusion. To address this, we introduced fluorescent dextran, a widely utilized tool for assessing BBB integrity [21], into the perfusate. Instead of using a single type of fluorescent dextran, we employed two types—FITC-Dextran and Texas Red-Dextran—sequentially during perfusion. This approach enabled us to differentiate between vessel dilation and BBB disruption, both of which can alter FITC-Dextran concentration in the brain. Our results indicated an 18-fold increase in the FITC/Texas Red ratio in the 2% NaNO_2_ group compared to the PBS group, suggesting BBB disruption during perfusion with 2% NaNO_2_.

As a result, we recommend avoiding the use of 2% NaNO_2_ for mouse perfusion. Nonetheless, there remains a need for a method that can more effectively clear blood from the brain than PBS perfusion alone. After evaluating various vasodilators at different concentrations, we identified that 100 mM NaNO_2_ could further reduce residual blood without affecting the FITC/Texas Red ratio compared to PBS perfusion. To confirm this finding, we perfused mice with Evans Blue solution followed by PBS. The presence of blue-dyed brains in the 2% NaNO_2_ group, but not in the PBS and 100 mM NaNO_2_, validated our hypothesis that 2% NaNO_2_ disrupts the BBB, whereas 100 mM NaNO_2_ does not.

When optimizing the perfusion procedure, we found that Ket/Xyl anesthesia is a superior method for inducing unconsciousness before perfusion compared to CO_2_ euthanasia. Ket/Xyl anesthesia reduces the formation of blood clots, which can obstruct the flow of the perfusate in the vessels, a common issue with CO_2_ euthanasia. Consequently, after PBS perfusion, mice euthanized with CO_2_ exhibited pink brains due to residual blood. In contrast, mice anesthetized with Ket/Xyl showed significantly lighter brain coloration, confirmed by lower hemoglobin levels in the brain. Additionally, we observed that hemoglobin levels decreased as the perfusion rate increased, with optimal results at 10 mL/min. However, increasing the perfusion rate to 20 mL/min resulted in intolerable pressure on the cardiovascular system, causing leakage of the perfusate from the circulatory system.

Following the optimization of the perfusion procedure, we implemented the Ket/Xyl anesthesia method and a 10 mL/min perfusion flow rate in subsequent experiments to examine the impact of 100 mM NaNO_2_ on the biodistribution of large molecules and the permeability of blood vessels. As anticipated, 100 mM NaNO_2_ significantly reduced hemoglobin concentrations in the brain and kidneys. However, similar reductions were not observed in the liver and spleen. The spleen, an organ responsible for filtering blood, storing red blood cells, and removing old or damaged red blood cells (RBCs) that cannot pass through the narrow inter-endothelial slits [22,23], retained a high concentration of hemoglobin. Despite the vasodilatory effects that facilitate blood removal from the spleen, the presence of both normal and phagocytized RBCs in the spleen contributes to its elevated hemoglobin levels. The liver also plays a role in the removal of old or damaged RBCs [24,25], but unlike the spleen, it does not store RBCs. RBCs in the liver are typically confined to the blood vessels, including the sinusoids [26]. Consequently, after perfusion, the liver’s hemoglobin concentration is lower than that of the spleen, despite both organs being involved in the clearance of unhealthy RBCs.

By perfusing with fluorescent dextrans and Evans Blue, we confirmed the integrity of the BBB during a 10-min perfusion with 100 mM NaNO_2_. Unlike the BBB, blood vessels in other tissues may lack tight junctions, potentially allowing dextran to penetrate these tissues. Nevertheless, the fluorescence levels and the FITC/Texas Red ratio can still be utilized to assess changes in blood vessel permeability across different tissues. Generally, FITC exhibits lower membrane permeability compared to Texas Red, which is more lipophilic. Consequently, the FITC/Texas Red ratio is typically below 1, as observed in brain, liver, and spleen tissues.

Following perfusion, the observed fluorescence in tissues resulted from both vascular and extravascular sources. Sodium nitrite-induced vasodilation increased the vascular fluorescence contribution for both FITC-Dextran and Texas Red-Dextran. Consequently, the concentrations of FITC-Dextran and Texas Red-Dextran were higher in the brain, liver, and spleen of the 100 mM NaNO_2_ group compared to the PBS group (Figure 6A,B,D,E,J,K). Additionally, the increase in perfusion volume in the tissue caused by the sodium nitrite-induced vasodilation can slightly increase the extravascular contribution of FITC-Dextran in many tissues, except in the brain, which is protected by the BBB. Therefore, there was a minor difference in the FITC/Texas Red ratios between groups for the liver and spleen (Figure 6F,L).

In contrast, the kidney, an organ responsible for dextran clearance, displayed different results. In the PBS group, the concentration of FITC-Dextran in the kidney was substantially higher than in other tissues (Figure 6A,D,G,J). Additionally, the FITC-Dextran concentration exceeded that of Texas Red-Dextran, resulting in an FITC/Texas Red ratio greater than one in the kidney (Figure 6G–I). According to Equation (1), this suggests that a significant amount of FITC-Dextran passed through the glomerular membrane into the Bowman’s capsule, thereby increasing the extravascular FITC-Dextran in the kidney. Interestingly, the increase in extravascular FITC-Dextran in the 100 mM NaNO_2_ group did not match the increase observed in other groups. Thus, despite NaNO_2_ perfusion raising vascular FITC-Dextran levels, the total FITC-Dextran concentration in the 100 mM NaNO_2_ group did not exceed that of the other groups (Figure 6G). Dextrans are not thought to be reabsorbed or secreted by the renal tubules [27,28]. Therefore, the observed effects might be due to sodium nitrite’s potential inhibition of glomerular filtration or facilitation of dextran transport from the kidney to the bladder.

Compared to the non-perfusion group, contamination from vascular hIgG1 and hTf was significantly reduced by transcardiac perfusion, regardless of whether vasodilators were present. The optimized perfusion procedure, involving Ket/Xyl anesthesia followed by a 10-min perfusion with heparinized PBS solution at 10 mL/min, effectively minimized blood contamination. There were no significant differences in the K_p,t_ of hIgG1 and hTf among the perfusion groups. The 100 mM NaNO_2_ group showed a slight advantage in the K_p,t_ of hIgG1 and hTf specifically for the kidney and spleen.

This study has several limitations. Note, our recommended perfusion method is only optimized in mice and is mainly focused on perfusion efficiency in the brain. Rather than measuring hemoglobin levels across all tissues, we focused on assessing hemoglobin and drug concentrations specifically in the brain and key organs, including the liver, spleen, and kidney. Consequently, we cannot conclusively state that 100 mM NaNO_2_ enhances perfusion performance across all tissues compared with PBS. While hemoglobin is a reliable marker for residual blood in the brain, it may not accurately represent residual blood in other tissues, such as the liver and spleen. Though we don’t expect it to behave inferiorly in other species, as the next step, the optimized brain perfusion method also needs to be tested in higher species, such as canines and non-human primates. While direct perfusion studies cannot be conducted in humans, our optimized perfusion method enables more accurate determination of drug concentrations in target tissues. This increased accuracy improves our ability to predict human pharmacokinetics, which is essential for selecting appropriate human doses and reducing the risk of toxicity due to overdose.

## 4. Materials and Methods

### 4.1. Chemicals and Reagents

Fluorescein isothiocyanate–dextran, average mol wt 40,000 (FITC-Dextran), sodium nitrite (NaNO_2_), and human transferrin were obtained from Sigma-Aldrich (St. Louis, MO, USA). Dextran, Texas Red™, 40,000 MW, Neutral (Texas Red-Dextran) was purchased from ThermoFisher (Eugene, OR, USA). Human IgG1 (hIgG1) was from Athens Research & Technology (Athens, GA, USA).

### 4.2. Animals

Male C57BL/6 mice were obtained from Charles River Laboratory (Wilmington, MA, USA). After arrival, the mice were acclimated for a minimum of 72 h in a controlled environment with a 12-h light/dark cycle, maintaining appropriate temperature and humidity, and with unrestricted access to food and water.

### 4.3. Transcardiac Perfusion to Evaluate the Influence of Vasodilators on Blood Removal and Vascular Permeability

To investigate the effectiveness of vasodilators in removing blood from the cerebral vasculature and their influence on vascular permeability, mice were perfused with PBS containing 20 IU/mL heparin and fluorescent dextrans (FITC-Dextran and Texas Red-Dextran) either with or without vasodilators. In each perfusion experiment, the mouse was immediately fixed to a Styrofoam board after being euthanized by CO_2_. The heart and lungs were fully exposed by opening the abdominal and thoracic cavities. After collecting blood from the right ventricle, a 25 G needle was inserted from the tip into the center of the left ventricle. A small incision was made in the right atrium, and the perfusate was pumped into the mouse at 5 mL/min for 20 min, as described in Figure 1B. Blood samples collected in tubes with a clotting activator were centrifuged at 8000× *g* for 5 min to prepare serum samples. Brain samples were collected and stored at −80 °C until analysis.

For verifying the BBB integrity during perfusion, mice were perfused with a 0.1 mg/mL Evans Blue solution containing 1% BSA in the absence or presence of vasodilators, followed by 2 min’ PBS perfusion to replace the Evans Blue in the vessels. Brain tissues were then collected and photographed.

### 4.4. Transcardiac Perfusion to Optimize Experimental Procedure

As described in Table 1, to optimize the experimental procedure, we induced unconsciousness in mice using either CO_2_ euthanasia or Ketamine/Xylazine (100/10 mg per kg) anesthesia. After induction, the mice were perfused with heparinized PBS containing fluorescent dextrans at a flow rate of 5 mL/min for 20 min. For optimizing the perfusion flow rate and osmotic pressure of the perfusate, mice were anesthetized with Ketamine/Xylazine, followed by perfusion with PBS or 100 mM NaNO_2_ at varying flow rates, perfusion times, and osmotic pressures (Table 1).

### 4.5. Influence of Transcardiac Perfusion on the Tissue Distribution Study of Large Molecules

Human IgG1 and Transferrin were administered intravenously to the mice at a dose of 10 mg/kg 24 h prior to the perfusion study. After anesthesia with Ket/Xyl and blood collection, the mice were perfused with heparinized PBS containing fluorescent dextrans at a flow rate of 10 mL/min for 10 min (Figure 1B). Post-perfusion, brain, liver, kidney, and spleen tissues were collected, weighed, and stored at −80 °C. A serum was prepared by centrifuging the blood samples at 8000× *g* for 5 min, which was stored at −80 °C until analysis.

### 4.6. Sample Preparation

Before analysis, tissue samples were homogenized in PBS at a ratio of 1 g of tissue to 4 mL of PBS. Brain samples were thoroughly homogenized for 1 min in Lysing Matrix D tubes (MP Biomedicals, Irvine, CA, USA). Liver, kidney, and spleen samples were pre-cut before being placed in Lysing Matrix S tubes. After adding cold PBS, these tissues were homogenized twice for 40 s, with a 1-min rest between homogenizations. The homogenates were then centrifuged at 2500× *g* for 20 min, and the supernatants were collected and stored at −80 °C until analysis.

### 4.7. Quantification of Hemoglobin and Fluorescent Dextrans

Hemoglobin concentrations were measured using mouse hemoglobin ELISA kits (ab157715, Abcam, Waltham, MA, USA). For quantifying fluorescent dextrans, supernatants from tissue homogenates of mice perfused with PBS alone, without fluorescent dextrans, served as background controls. Calibration standard curves were prepared in a 1% BSA solution. A 100 µL aliquot of the diluted sample was transferred to a clear 96-well plate for reading with a plate reader (Molecular Devices SpectraMax M5, San Jose, CA, USA). The excitation, emission, and cutoff values were set to 485, 525, and 515 nm for FITC-Dextran and 586, 625, and 610 nm for Texas Red-Dextran.

### 4.8. Quantification of Human IgG1 and Human Transferrin

The concentrations of human IgG1 (hIgG1) in serum and tissues were quantified using an electrochemiluminescence assay. Briefly, 100 µL of diluted samples in 1% BSA in PBS (pH 7.4) were added to a 96-well plate coated with mouse anti-human Fc CH3 domain-specific antibody (Fisher Scientific, Eugene, OR, USA, CAT # MA5-16557). After a 2-h incubation at room temperature (RT), the plate was washed. Subsequently, 100 µL of a detection reagent (0.5 µg/mL anti-human antibody (Goat) Sulfo-TAG labeled in blocking buffer) was added and incubated for 40–45 min at RT with shaking. Following another wash step, 150 µL of an MSD reading buffer was added, and the plate was read immediately using the MESO QuickPlex SQ120 (Rockville, MD, USA).

Human transferrin (hTf) concentrations were measured by ELISA. The ELISA procedure was similar to the MSD assay. The Human Transferrin Antibody Pair (Abcam, Cat# ab253565) was used, with the detector antibody conjugated to HRP using the HRP Conjugation Kit (Abcam, Cat# ab102890) prior to use. Concentrations were calculated using GraphPad Prism (version 9).

### 4.9. K_p,b_ and K_p,t_ Determination and Statistics

The brain/serum (K_p,b_) and tissue/serum (K_p,t_) ratios were calculated by dividing the drug concentrations in the brain and tissues by the serum drug concentration, respectively. Statistical comparisons between groups were performed using an unpaired Student’s *t*-test, while comparisons among multiple groups were conducted using one-way ANOVA with post hoc Dunnett’s or Tukey’s test, all analyzed with GraphPad Prism (version 9). A *p*-value of <0.05 was considered statistically significant.

## 5. Conclusions

In conclusion, we have developed an effective method for evaluating BBB integrity during transcardiac perfusion. Due to minimal interference between FITC-Dextran and Texas Red-Dextran, both fluorescent dextrans can be simultaneously perfused and quantified in mice. This approach allows for the assessment of vascular permeability not only in brain tissue but also in other tissues by comparing the FITC-Dextran to Texas Red-Dextran ratio between the vasodilator and PBS groups. Our optimization of the perfusion procedure revealed that Ket/Xyl anesthesia is superior to CO_2_ euthanasia, and a perfusion rate of 10 mL/min is optimal. The use of heparinized PBS solution with 100 mM NaNO_2_ effectively removes residual blood from tissues, resulting in lower K_p,t_ values for hIgG1 and hTf in specific tissues without significantly affecting blood vessel permeability.

## Figures and Tables

**Figure 1 ijms-25-12180-f001:**
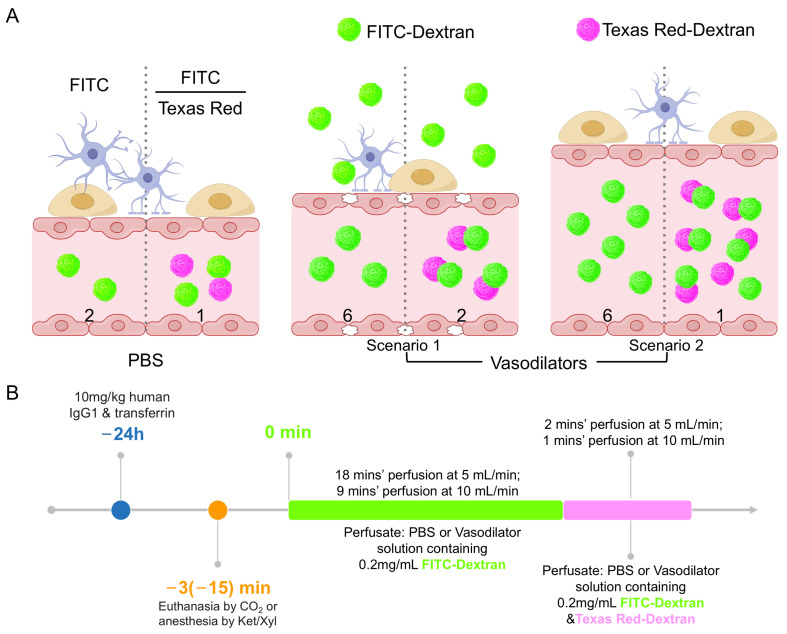
Schematic diagram of study design and rationales by Figdraw. (**A**) Rationale for using fluorescent dextran to determine BBB integrity. The situations of single fluorescent (FITC) dextran perfusion and dual fluorescent (FITC & Texas Red) dextran perfusion are shown on the left and right side of the dotted line, respectively. The numbers in the vessels represent the FITC concentration (left side of dotted line) and the FITC-Dextran to Texas Red-Dextran ratio (right side). (**B**) The timeline of the in vivo experiment.

**Figure 2 ijms-25-12180-f002:**
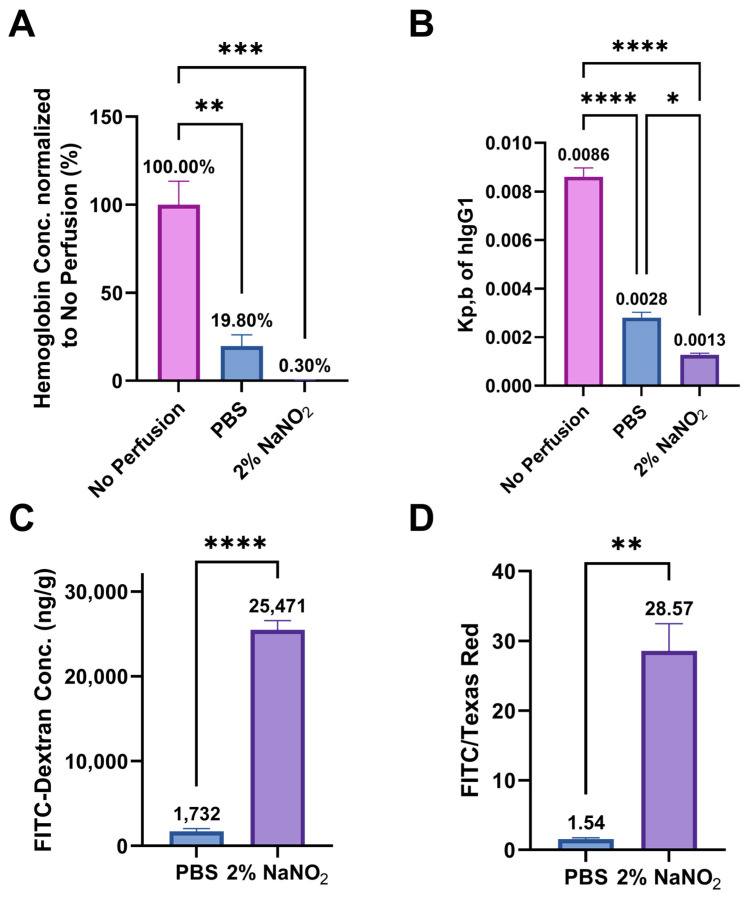
Performance of 2% NaNO_2_ perfusion (*n* = 3). (**A**) Hemoglobin concentration of mice perfused with PBS or 2% sodium nitrite after CO_2_ euthanasia. Hemoglobin concentration in each group was normalized to that of the no perfusion group. (**B**) K_p,b_ of human IgG1, (**C**) FITC-Dextran concentration in the brain tissue, and (**D**) the ration of FITC-Dextran to Texas Red-Dextran after PBS and 2% NaNO_2_ perfusion. * Represents significant differences (*p* < 0.05); ** is *p* < 0.01, *** is *p* < 0.001; and **** is *p* < 0.0001.

**Figure 3 ijms-25-12180-f003:**
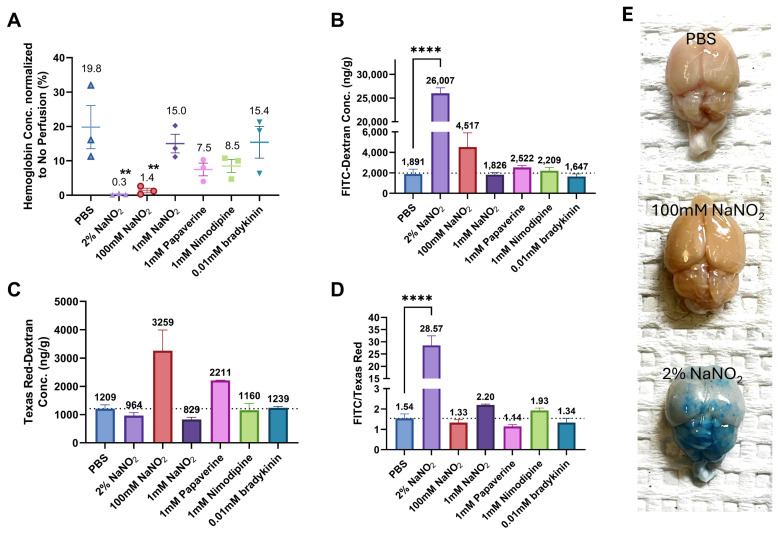
Evaluation of the performance of vasodilators’ perfusion (*n* = 3). (**A**) Hemoglobin concentration, (**B**) FITC-Dextran concentration, (**C**) Texas Red-Dextran concentration, and (**D**) the FITC/Texas Red ratio in the brain tissue were measured and analyzed for each perfusion group as mice were euthanized by CO_2_. The superscripted ** in (**A**) represent significant differences (*p* < 0.01) compared with the PBS group (one-way ANOVA with post hoc Dunnett’s test). Except for the 2% NaNO_2_ group, no significant difference in the FITC/Texas Red ratio was observed between PBS and other groups (one-way ANOVA with post hoc Dunnett’s test *t*). To verify the impact of NaNO_2_ perfusion on BBB integrity, mice were perfused by the Evans Blue in 1% BSA solution followed by 2-min PBS perfusion. (**E**) The photograph of brain tissue was captured immediately after perfusion. **** represents significant differences (*p* < 0.0001).

**Figure 4 ijms-25-12180-f004:**
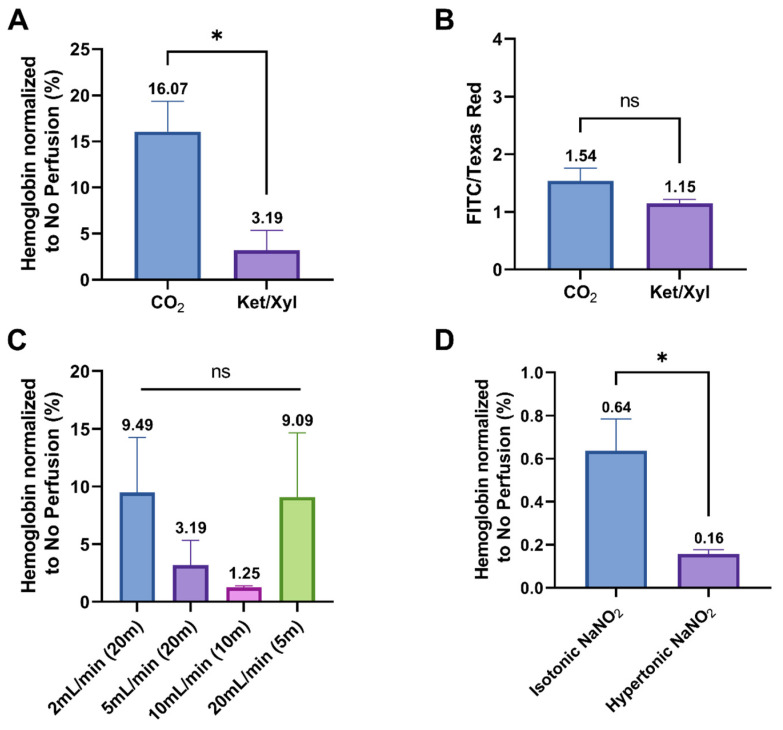
Optimization of perfusion procedure including (**A**,**B**) induction of unconsciousness, (**C**) perfusion flow rate, and (**D**) osmotic pressure. (**A**,**B**) Before PBS perfusion, compared to CO_2_ euthanasia, using Ket/Xyl to anesthetize the mouse significantly reduced hemoglobin concentration in the brain without changing the FITC/Texas Red ratio (unpaired Student’s *t*-test). (**C**) The PBS perfusion decreased hemoglobin concentration to the lowest level when perfusion rate was 10 mL/min, although there was no significant difference in the hemoglobin concentration among groups (one-way ANOVA). (**D**) The 100 mM NaNO2 in PBS solution (hypertonic, 463 mOsmol/L) removed more hemoglobin from the brain vessels (unpaired Student’s *t*-test). * Represents significant differences (*p* < 0.05). The “ns” is an abbreviation for “not significant”.

**Figure 5 ijms-25-12180-f005:**
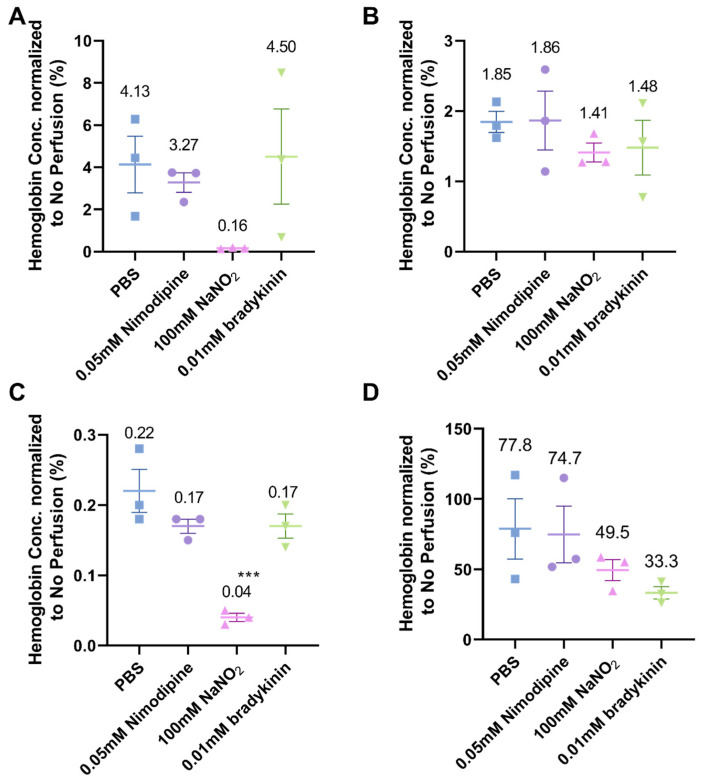
Hemoglobin concentration in different tissues (*n* = 3), including (**A**) brain, (**B**) liver, (**C**) kidney, and (**D**) spleen. Mice were anesthetized by Ket/Xyl and perfused by different solutions at 10 mL/min in the experiment. (**A**–**C**) The 100 mM NaNO_2_ was the most effective at reducing hemoglobin concentration in brain, liver and kidney. (**D**) In the spleen, the hemoglobin concentration was extremely high in all groups (33.3–77.8%). The superscripted *** in (**C**) represent significant differences (*p* < 0.001) compared with the PBS group.

**Figure 6 ijms-25-12180-f006:**
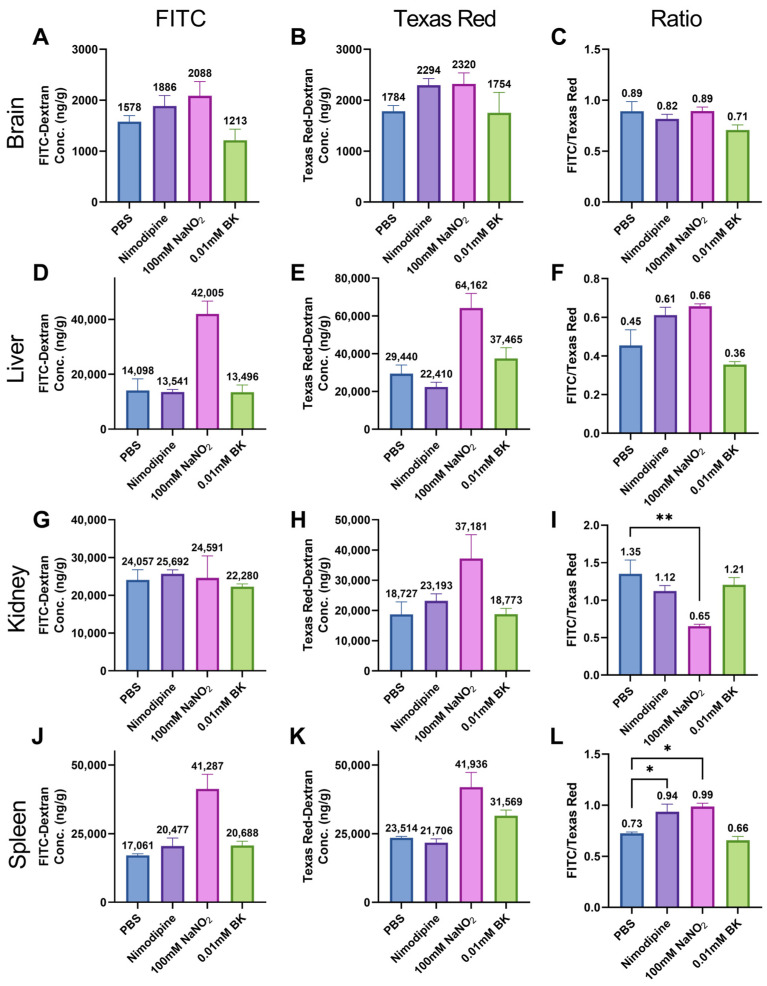
Fluorescent dextran concentrations in different tissues (*n* = 3). Compared to PBS group, 100 mM NaNO_2_ increased FITC-Dextran concentration in the (**A**) brain, (**D**) liver, and (**J**) spleen, but not in the (**G**) kidney. Regarding Texas Red-Dextran, 100 mM NaNO_2_ increased its concentration in all tissues (**B**,**E**,**H**,**K**). As a result, there was minor difference in FITC/Texas Red ratio between groups in the (**C**) brain and (**F**) liver. However, in the (**I**) kidney and (**L**) spleen, a significant decrease in 100 mM NaNO_2_ group was observed compared with PBS group. (One-way ANOVA with post hoc Tukey’s test.) * Represents significant differences (*p* < 0.05); and ** is *p* < 0.01.

**Figure 7 ijms-25-12180-f007:**
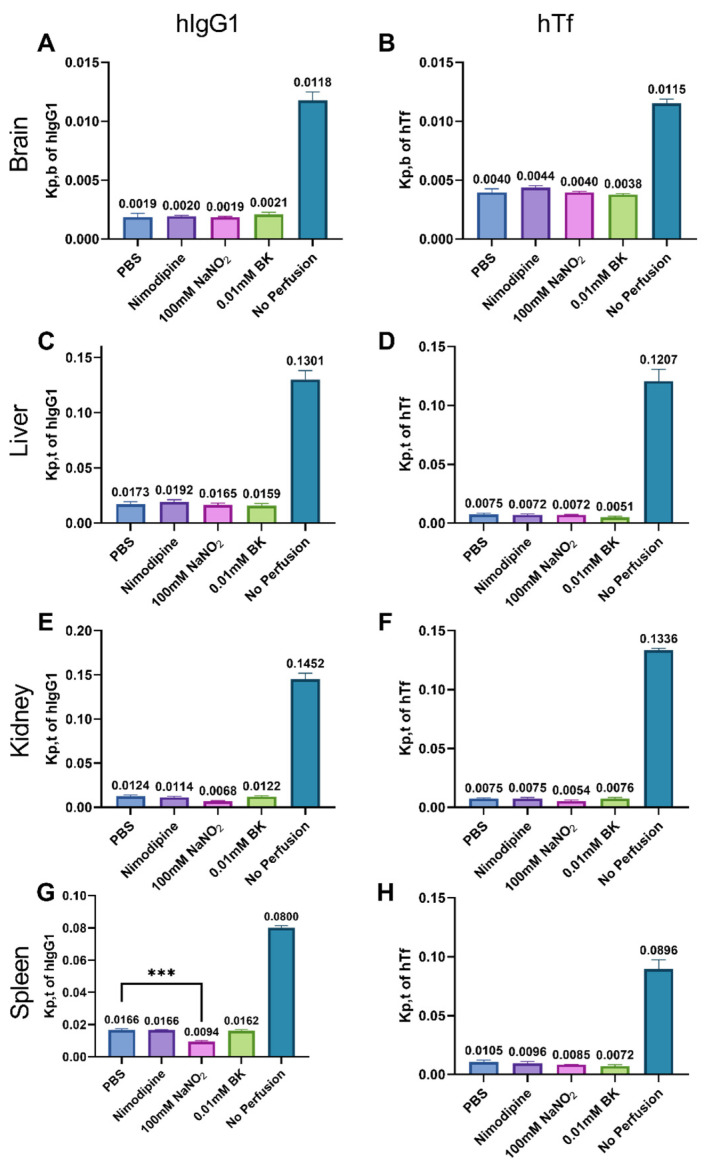
K_p,b_ and K_p,t_ of human IgG1 and transferrin (**A**–**D**). Regarding K_p,b_ and K_p,t_ of hIgG1 and hTf in the brain and liver, perfusion reduced the K_p,b_ and K_p,t_ value in comparison of no perfusion group. However, no difference was found between perfusion groups (**E**,**F**). For the kidney, compared to PBS perfusion, 100 mM NaNO_2_ reduced K_p,t_ of hIgG1 and hTf without significance. (**G**,**H**) 100 mM NaNO_2_ significantly lowered the K_p,t_ of hIgG1 in the spleen but not for K_p,t_ of hTf compared with PBS group (one-way ANOVA with post hoc Dunnett’s test). *** Represents significant differences (*p* < 0.001).

**Table 1 ijms-25-12180-t001:** Perfusion experiment conditions.

Induction of Unconsciousness	Flow Rate (mL/min)	Perfusion Time (min)	NaNO_2_ Concentration (mM)	Osmotic Pressure (mOsmol/L)
CO_2_	5	20	0	283
Ket/Xyl	5	20	0	283
	2	20	0	283
	5	20	0	283
	10	10	0	283
	20	5	0	283
	10	10	100	463
	10	10	100	283

## Data Availability

Data is contained within the article.

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
