# Peer review of "Optimization of Transcardiac Perfusion for More Accurately Evaluating Biodistribution of Large Molecules"

_ijms, 2024, doi:10.3390/ijms252212180_

Round 1
Reviewer 1 Report
Comments and Suggestions for Authors
The study focuses on the effects of different vasodilators, including sodium nitrite (NaNO2), on the integrity of the blood-brain barrier and the removal of leftover blood from tissues after perfusion. The effects of different concentrations and perfusion settings, such as flow rate and anesthetic type (CO2 versus Ketamine/Xylazine), on NaNO2 were assessed. FITC-Dextran and Texas Red-Dextran, two bright markers, made it possible to analyze BBB permeability.
The work discusses a vital topic in the study of medication biodistribution, specifically in relation to the examination of biological components in the brain. This is important for developing treatments that must cross the blood-brain barrier, such those for neurological illnesses.
It is novel and offers a more accurate evaluation to differentiate between vasodilation and BBB disruption using two types of fluorescent dextran (FITC and Texas Red).
Future studies on biodistribution can directly benefit from the discovery that 100 mM NaNO2 is an effective concentration for removing leftover blood without jeopardizing the integrity of the blood-brain barrier.
The text should be improved at some points:
-While the study primarily focused on the brain, it would be interesting to do a more thorough review of how the perfusion approach may be enhanced for other organs, as different tissues—like the liver and spleen—removed blood differently.
- Although more testing in other species (dogs, non-human primates) is mentioned in the study, it would be helpful to have a more thorough description of how the findings in this model may be applied to humans.
-To make the data simpler to understand, several graphs and tables—particularly those that compare the various experimental groups—might be better arranged.
To improve the article, especially in the introduction and discussion, research that address perfusion techniques, evaluation of the integrity of the blood-brain barrier (BBB), and biodistribution of big molecules should be added.
One suggestion:Newly Approved Studies of Brain-Penetrating Biologics:
Mullard, A. (2021). Blood-brain barrier-traversing biologic secures regulatory approval, in Japan. Nature Reviews Drug Discovery. This article covers recent advances in BBB-crossing biologics, which may contextualize the clinical relevance of their study.
Author Response
Dear Reviewer,
Thank you very much for your insightful comments, which will help enhance the quality and clarity of our manuscript. Below are our responses to each point.
Point-by-point response to Comments and Suggestions
Comments 1:
-While the study primarily focused on the brain, it would be interesting to do a more thorough review of how the perfusion approach may be enhanced for other organs, as different tissues—like the liver and spleen—removed blood differently.
Response 1:
We completely agree that exploring perfusion performance in other organs is a valuable area of study. For this reason, in addition to the brain, we also examined how perfusion impacts hemoglobin and biologic concentrations in organs like the liver, kidney, and spleen. Our findings revealed some intriguing observations that merit further investigation
Comments 2:
- Although more testing in other species (dogs, non-human primates) is mentioned in the study, it would be helpful to have a more thorough description of how the findings in this model may be applied to humans.
Response 2:
Thank you for emphasizing the need for a deeper discussion on the potential application of our findings to humans. We add following discussion in the last paragraph in the Discussion section. “While direct perfusion studies cannot be conducted in humans, our optimized perfusion method enables more accurate determination of drug concentrations in target tissues. This increased accuracy improves our ability to predict human pharmacokinetics, which is essential for selecting appropriate human doses and reducing the risk of toxicity due to overdose.”
Comments 3:
To make the data simpler to understand, several graphs and tables—particularly those that compare the various experimental groups—might be better arranged.
Response 3:
Thank you for your suggestion. To improve readability, we have reorganized the Figure 3 which contain many experimental groups to better illustrate comparisons across experimental groups
Comments 4:
To improve the article, especially in the introduction and discussion, research that address perfusion techniques, evaluation of the integrity of the blood-brain barrier (BBB), and biodistribution of big molecules should be added.
One suggestion:Newly Approved Studies of Brain-Penetrating Biologics:
Mullard, A. (2021). Blood-brain barrier-traversing biologic secures regulatory approval, in Japan. Nature Reviews Drug Discovery. This article covers recent advances in BBB-crossing biologics, which may contextualize the clinical relevance of their study.
Response 4:
Thank you for the suggestion to include Mullard’s 2021 article on BBB-penetrating biologics. We already cited it (reference 20) in the discussion section to underscore the clinical significance of blood-brain barrier research and to frame our study within the context of ongoing advancements in BBB-crossing therapeutics.
We hope these revisions address your valuable feedback and look forward to any further comments.
Reviewer 2 Report
Comments and Suggestions for Authors
The research work presented in the article titled “Optimization of transcardiac perfusion for accurately evaluating biodistribution of large molecules” is an appreciable attempt.
General comments:
1. The data has been presented systematically though; still some more clear explanation of the procedures is needed.
2. More optimizations needed to claim the finding as ‘accurate evaluation’
3. A few English corrections Line 142, present (it should be presence) Line 217, ration (it should be ratio)
Queries:
Q1. Regarding Lines 319, 320
Compared to no perfusion, all perfusion groups significantly reduced hIgG1 and hTf concentrations in all tissues.
How did authors prepare samples from the tissues for these experiments? Meaning, the solutions to be measured in electrochemiluminescence assay or ELISA) for obtaining the concentrations of hIgG1 and hTf. This is not clearly reflected in the manuscript.
This sample acquisition is crucial in these experiments. After all, the result obtained from the assays is reduced hIgG1 and hTf concentrations, which could be a false positive also.
Q2. What is the strength of the PBS used for the studies?
Q3. In the title, “………. for accurately evaluating biodistribution” It might be a catchy title, but not justifiable based on the study presented here and its limitations.
Q4. Fluorescence technique is potential though, e.g., imaging but for quantification in this context could be sometimes misleading. As it depends on various physicochemical factors.

Author Response
|
Comments 1: Q1. Regarding Lines 319, 320 Compared to no perfusion, all perfusion groups significantly reduced hIgG1 and hTf concentrations in all tissues. How did authors prepare samples from the tissues for these experiments? Meaning, the solutions to be measured in electrochemiluminescence assay or ELISA) for obtaining the concentrations of hIgG1 and hTf. This is not clearly reflected in the manuscript. This sample acquisition is crucial in these experiments. After all, the result obtained from the assays is reduced hIgG1 and hTf concentrations, which could be a false positive also. |
|
Response 1: Thank you for pointing this out. We agree with this comment. Therefore, we have added details about the sample preparation in the method section. 4.6. Sample preparation Before analysis, tissue samples were homogenized in PBS at a ratio of 1 g of tissue to 4 mL of PBS. Brain samples were thoroughly homogenized for 1 minute in Lysing Matrix D tubes (MP Biomedicals, Irvine, CA). Liver, kidney, and spleen samples were pre-cut before being placed in Lysing Matrix S tubes. After adding cold PBS, these tissues were homogenized twice for 40 seconds, with a 1-minute rest between homogenizations. The homogenates were then centrifuged at 2,500 x g for 20 minutes, and the supernatants were collected and stored at -80°C until analysis. |
|
Comments 2: Q2. What is the strength of the PBS used for the studies? |
|
Response 2: Thank you for the question. We used 1xPBS in the study.
|
|
Comments 3: Q3. In the title, “………. for accurately evaluating biodistribution” |
|
Response 3: Thank you for pointing this out. We have adjusted the title to avoid an absolute tone. This is the new title: Optimization of transcardiac perfusion for more accurately evaluating biodistribution of large molecules in rodents
|
|
Comments 4: Fluorescence technique is potential though, e.g., imaging but for quantification in this context could be sometimes misleading. As it depends on various physicochemical factors. |
|
Response 4: Thank you for pointing this out. We agree that some fluorescence intensity could be influenced by molecular conditions. To address this, we selected commonly used fluorescent dextran and tested for potential interference and matrix effects before applying it in perfusion |
We hope that these revisions meet your expectations and enhance the quality of our manuscript. Thank you again for your valuable feedback and guidance.